# Implication of Freeze–Thaw Erosion and Mechanism Analysis of High-Performance Aromatic Liquid Crystal Fibers

**DOI:** 10.3390/polym15092001

**Published:** 2023-04-23

**Authors:** Hai Wan, Yanping Wang, Wenbin Jin, Shuohan Huang, Yimin Wang, Yong He, Peng Wei, Yuwei Chen, Yumin Xia

**Affiliations:** 1State Key Laboratory for Modification of Chemical Fibers and Polymer Material, College of Materials Science and Engineering, Donghua University, Shanghai 201620, China; 2Key Laboratory of High Performance Fibers & Products, Engineering Research Center of Technical Textiles, College of Science, Donghua University, Ministry of Education, Shanghai 201620, China; 3Henan Key Laboratory of Functional Textile Materials, College of Textiles, Zhongyuan University of Technology, Zhengzhou 450007, China; 4Key Laboratory of Rubber-Plastics, Ministry of Education/Shandong Provincial Key Laboratory of Rubber-Plastics, Qingdao University of Science & Technology, Qingdao 266042, China

**Keywords:** high-performance liquid crystal fiber, PPTA fiber, TLCP fiber, freeze–thaw experiment, ion penetration, microfibrillar structure

## Abstract

According to the demand for high-performance fibers for high-latitude ocean exploration and development, this paper selects representative products of high-performance liquid crystal fibers: thermotropic liquid crystal polymer fibers (TLCP) and poly p-phenylene terephthalamide (PPTA) fibers. Through a series of freeze–thaw (F–T) experiments for simulating a real, cold marine environment, we then measure the retention of mechanical properties of these two kinds of fibers. Before that, due to the difference in their chemical structures, we tested their Yang–Laplace contact angle (YLCA) and water absorption; the results suggested that PPTA fibers would absorb more moisture. Surprisingly, then, compared with thermotropic liquid crystal polymer (TLCP) fibers, the retention of the mechanical properties of poly p-phenylene terephthalamide (PPTA) fibers decreased by around 25% after the F–T experiments. The Fourier-transformed infrared (FT-IR) analysis, the attenuated total reflection (ATR) accessory analysis and the degree of crystal orientation measured by two-dimensional wide-angle X-ray diffraction (2D-WAXD) confirm that no changes in the chemical and the orientation structure of the crystal region of the fibers occurred after they underwent the F–T cycles. However, as observed by scanning electron microscopy (SEM), there are microcracks of various extents on the surface of the PPTA fibers, but they do not appear on the surface of TLCP fibers. It is obvious that these microcracks will lead to the loss of mechanical properties; we infer that the moisture absorbed by the PPTA fibers freezes below the freezing point, and the volume expansion of the ice causes the collapse of the microfibrillar structure. The two sorts of fibers subjected to the F–T experiments are immersed in a sodium chloride solution, and the amount of water infiltrated into the PPTA microfibrillar structure is evaluated according to the content of sodium ions in the fiber surface and subsurface layers through X-ray spectroscopy (EDS) elemental analysis. From the above analysis, we found that TLCP fibers can more effectively meet the operating standards of the severe and cold marine environment.

## 1. Introduction

More than half a century after the discovery of liquid crystals, liquid crystal polymer materials have ushered in an era of great development. In 1965, DuPont invented Kelvar^®^, a kind of lyotropic liquid crystal polyaromatic amide which produced PPTA fibers by solution spinning. It is a type of high-tensile strength and strongly heat-resistant liquid crystal fiber, and a representative of lyotropic liquid crystal polymers [1]. In the 1970s, with the rise of the electronics industry, the demand for high-precision injection molding parts increased, which prompted the expansion of thermotropic liquid crystal polymers. However, aromatic polyester, which forms polymer materials through linking to aromatic rings with ester bonds, is the representative of those polymers. In 1976, Hoechst Celanese introduced the naphthalene ring structure into the main chain of aromatic polyester to develop the wholly aromatic thermotropic liquid crystal polyacrylate Vectra^®^, which is a landmark product. Due to its outstanding spinnability, in the 1990s, Kuraray, in Japan, worked with Celanese to develop the liquid crystal polyacrylate fiber Vectran^®^, and made it the only commercialized thermotropic liquid crystal fiber in the world [2]. Compared with those aliphatic polyamide and polyester fibers, the advantages of the wholly aromatic polyamide and polyester are mainly reflected in the properties of high tensile strength, a high initial modulus, good impact resistance and splendid high-temperature resistance, which give them a wider range of application [3,4,5].

In recent years, this high-performance fiber has played key role in making up for performance shortcomings for rope applications in the marine and military industries, ranging from lifting to temporary and permanent mooring systems for vessels and offshore oil–gas drilling platforms. Selecting a rope made by high-performance fiber brings several excellent advantages over traditional steel ropes, such as a high load-to-weight ratio, good fatigue resistance and safety, that perhaps make them an outstanding candidate to replace steel ropes [6]. For example, the growth in demand for petroleum has led to the exploration of new reserves in deeper and deeper areas of the ocean, sometimes involving prospection depths of more than 1000 m. It is necessary to develop mooring systems that allow lighter and stiffer lines to connect floating units, which are constantly subject to the influence of the ocean environment (e.g., waves, wind and currents), with the ocean floor. Therefore, mooring systems that aim to restrict the position of the floating units thus provide operational safety for the platforms, which will be guaranteed by the anchoring line. This equipment, which is the core component of the system, will require the ropes to be synthetic fibers that possess greater tenacity [7]. For some time now, studies have been conducted on synthetic fibers for deep-water mooring systems, but they are often limited to mechanical properties such as creep damage or fatigue damage, etc. [6,8]. When it comes to exploration, mooring and transportation in high-latitude marine areas, ropes made with high-performance fibers as a key component face the challenges of extreme low temperatures and the underwater wet environment, which test the excellent mechanical properties they already have. Clothes for scientific exploration in the Antarctic region have recently been studied [9]. These clothes are expected to withstand harsh conditions, and they must have certain functions to ensure the safety and comfort of scientists. This study inspired the current work.

The presence of water has an obvious effect on polyamide materials; on the one hand, the amide group will form hydrogen bonds between molecular chains, thereby increasing the intermolecular cohesion and improving its mechanical properties. On the other hand, it will bring about hydrophilicity of fabrics, resins and film materials [10]. The relationship between the stretching behavior of polyamide(PA) 6 film and water content has been studied by predecessors; the plasticization of water will change the mobility of molecular chains in the amorphous phase, and it will also penetrate into defective crystals, dismantling hydrogen bonds and breaking crystals, which have a decisive effect on the stress–strain response [11]. Some researchers have also studied the initial modulus of polyamide injection molding materials after moisture absorption. Since moisture has the effect of reducing the glass transition temperature of the polymer, the difference between the modulus in dry and wet states is more than 80% [12]. The effect of moisture absorption on the mechanical properties of glass fiber-reinforced polyamide composites has been explored [13]. Some scholars have studied the effect of moisture absorption on the mechanical properties of carbon nanotube-reinforced PA6 fiber composites [14].

In addition to aliphatic polyamide fiber, the accelerated creep properties of Kevlar^®^ after moisture absorption have also been demonstrated; load cycling magnifies accelerated creep with increasing moisture content [15]. Kevlar^®^ solution spinning contains a small amount of sulfuric acid solvent, which causes the finished fiber to be catalyzed and hydrolyzed under high humidity, resulting in partial degradation and loss of mechanical properties [16]. The moisture regains of aliphatic polyester fiber and polyamide fiber have been measured by predecessors at 0.4% and 4.5%, respectively, indicating that the hydrophilicity of ester groups is much lower than that of amide groups [17]. For wholly aromatic polyester and polyamide fibers, the moisture regain results were less than 0.1% and 4–6%, respectively, results which are consistent with the two aliphatic types of fibers [3].

Apart from the influence of moisture, when the ambient temperature is below zero, water undergoes a phase transition. When it is liquid, hydrogen bonds around the water molecules are continuously disassembled and reformed in a disordered manner. During the freezing process, water molecules take a more defined shape and arrange themselves in six-sided crystalline structures. On the contrary, it could result in more expansion. This means that the crystalline arrangement is less dense than that of the molecules in liquid form, which makes the ice less dense than the liquid water; thus, the volume expands by approximately 9% [18]. This effect has a disruptive effect on the structure of fiber-reinforced composites. One publication pointed out that increasing freeze–thaw cycles reduced the compressive strength of lime-stabilized basalt and polypropylene fiber-reinforced clays [19]. Lina Zhang [20] and her colleagues exploited the properties of volume change after freeze–thaw experiments and developed a series of mixed solvents of LiOH/urea and NaOH/urea to dissolve cellulose at −12 °C. The alkali “hydrate” is more easily attracted to the cellulose chain by forming a new hydrogen bond network which is more stable at low temperature, thereby dismantling the closely packed cellulose molecular chain structure with many hydrogen bonds and leading to its dissolution.

Therefore, the different effects of the presence of water on polyamide fiber and polyester fiber originate from the hydrophilic properties of their amide and ester functional groups, and combined with the change of ambient temperature, they lead to the freezing of water, which brings about volume expansion. In the course of practical application, how will this phenomenon affect the mechanical properties of high-performance fibers for high-latitude marine operations? Unfortunately, compared with heat resistance, chemical and UV resistance research into high-performance liquid crystal fibers, there is a lack of literature to provide sufficient data to prove the stability of the mechanical properties of PPTA and TLCP fibers, as representative products of high-performance fibers, in frequent freeze–thaw cycles. It is possible that this is the first study on the effect of ordinary water and temperature changes on the mechanical properties of high-performance liquid crystal fibers.

In this paper, first of all, based on the fiber hydrophilicity statement described above, we design a new experiment for measuring YLCA and moisture content to further prove that under the same experimental conditions, the discrepancy in the chemical structure of the polymers will endow the two types of liquid crystal fibers with completely different ratios of water absorption. In the next step, we utilize F–T cycle experiments to simulate the actual cold marine operating environment and obtain the retention rate of the mechanical properties of two sorts of high-performance fibers. According to the data of ATR-FTIR and 2D-WAXD, the variation in the chemical structure and the degree of crystal orientation of the kinds of fibers will be discussed in detail. The morphology of the liquid crystal fiber surface after multiple F–T cycles is observed using SEM; thus, the YLCA of these fibers has also been tested at the same time, and we hope to substantiate the damage to the fibers’ surfaces after the F–T experiments using the two different methods. The EDS elemental analysis will reveal that water molecules can penetrate into the microfibrillar structure of liquid crystal fibers under the F–T experiments through immersing the fibers in the same content of sodium chloride solution; thus, we intend to demonstrate the extent of damage to the surface or even subsurface of the fibers after the F–T experiments from another angle, which will finally affect the mechanical properties.

## 2. Experimental

### 2.1. Materials

There are two types of fibers used in the F–T experiments: the first are TLCP fibers, provided by Ningbo Higlar New Material Technology Co., Ltd. (Ningbo, China). The second are PPTA fibers, provided by Yantai Tayho Advanced Materials Co., Ltd. (Yantai, China). The two kinds of fibers are extracted and soaked with acetone and petroleum ether before further experiments in order to remove spinning oil or impurities from the surface. Their chemical structures are shown in Figure 1, and the tensile strength and initial modulus of the PPTA and TLCP fibers are 17.47 cN/dtex, 545.84 cN/dtex; 21.34 cN/dtex, and 722.86 cN/dtex, respectively. The diameter of the fibers is about 20 μm. Sodium chloride (AR ≥ 99.5%) was obtained from Shanghai Lingfeng Chemical Reagent Co., Ltd. (Shanghai, China). It was then made into a 10% solution with deionized water.

### 2.2. The Designed F–T Experiments

The two kinds of fibers are first soaked in deionized water for 24 h. Then, we designed experiments, as shown Table 1, to simulate the real environment. These samples can be used to characterize the retention of mechanical properties and surface morphology of the fibers after F–T experiments; we may use sample No. 3 to further study the extent of damage within the surface and the subsurface of the fibers through the content of sodium ions measured by EDS elemental analysis. This analysis occurs after the fibers are immersed in the same concentration of sodium chloride solution and dried.

### 2.3. Characterization Methods

#### 2.3.1. Tensile Strength Test

The mechanical properties of the high-performance fibers were measured by microcomputer control electronic universal testing machines, with load cells of 1 kN (JiNan MTS Test Technology, Jinan, China), according to GB/T19975-2005. The gauge length and crosshead speed were 500 mm and 250 mm·min^−1^, respectively. More than ten tensile test measurements were conducted for each sample.

#### 2.3.2. The 2D-WAXD Analysis

The degree of crystal orientation of two types of fibers were measured by the 2D-WAXD apparatus (D/max-2550VB+/PC, 18 kW, Rigaku, Japan) employing Cu Kα (λ = 1.54056 Å).

The degree of crystal orientation *R* is defined by Equation (1):(1)$$R=\frac{180-W}{W}\times 100\%$$

*W* is the half-width of the intensity distribution curve along the Debye–Scherrer ring.

#### 2.3.3. The FT-IR Spectra Analysis

The FT-IR spectra were taken on a Nicolet™ iS50 FTIR spectrometer (ThermoFisher, Waltham, MA, USA), with a spectral range of 525 to 4000 cm^−1^ and a built-in mid- and far-IR-capable diamond ATR accessory. To obtain the spectra, 32 scans were collected at a resolution of 4 cm^−1^.

#### 2.3.4. The Characterization of the Young–Laplace Contact Angle

Wettability is often characterized using the Young–Laplace contact angle (YLCA) by placing a small droplet on a solid surface and measuring the angle between the tangent to the droplet at the solid–liquid–air contact line and the surface [21]. Firstly, we fixed both ends of PPTA and TLCP monofilament yarns, suspended them in midair, and used a water jet to spray fine water mist on the top. The water mist fell and stayed on the monofilament yarns. Due to the action of gravity, the water forms droplets hanging on the monofilament yarns, as shown in Figure 2.

At this time, the Horizon HD-Resolution Flatness Detector (Kunshan Gaoping Instrument, Kunshan, China) was used to shoot, according to the schematic diagram in the figure. For yarns whose edges are not wetted, take the angle between the tangent of the droplet and the yarns; for yarns whose edges are wetted, take the angle between the tangent point of the outermost droplet under the wetted part and the fibers, and calculate the angle value using the software provided with the instrument.

#### 2.3.5. The Moisture Content Measurement

This experiment uses a different moisture content measurement method from previous works. Drying fibers need to be equilibrated for a long time in a constant humidity and temperature environment and then weighed [22]. However, this is not suitable for the working environment of high-performance fibers; in the cold marine field, the fibers need to be completely wetted by water and then frozen in the air, repeatedly. Therefore, our experimental method adopts the following steps: first, select several copies of two sorts of fibers of a certain quality, put them into a blast oven for drying, weigh them, then record the quality of fibers after drying for 6 h; second, immerse them in deionized water for 24 h, make sure the water and fibers fully contact each other, and then drain the excess water from the fibers, put them into a high-speed rotating centrifugal instrument (Henan Bei Hong TD5D, Zhengzhou, China), and rotate it them to further remove as much of the water that seeps into the surface of the microfibril as possible in the two kinds of liquid crystal fibers. Set the speed to 2000 rpm, and the duration to 4 min, and repeat 11 times.

#### 2.3.6. The SEM Measurement

The surface topographies of two sorts of high-performance liquid fibers were characterized using an environmental scanning electron microscope (ESEM-QUANTA 250, FEI, Brno, Czech Republic) operating at 12.5 kV and under high vacuum.

#### 2.3.7. The EDS Elemental Analysis

The energy disperse X-ray spectroscopy (EDS) analyses were performed on gold-powder coated yarns, which have been immersed in the sodium chloride solution and dried afterwards in an ESEM-QUANTA 250 ESEM (Brno, Czech Republic) coupled with an Aztec-X-Max 20 detector EDS system (Oxford Instruments, Abingdon, UK).

## 3. Results and Discussion

### 3.1. The Analysis of the YLCA and Moisture Content of Two Sorts of Fibers

Measurement of the liquid contact angle of fibers in some of the literature [23] is generally based on fabrics or bundles of many fibers, but the dense fibers’ arrangement will accelerate the diffusion of droplets on the fibers, for example, through the capillary phenomenon [24], which causes the droplets to be absorbed rapidly by the fabric, meaning relatively accurate test results cannot be achieved. Indeed, due to the elongated cylindrical geometry of the fibers, the wettability of the fibrous material is quite different from that of the same material in the form of a flat surface [25]. Therefore, in the absence of a universally accepted method for measuring the droplet contact angle on a fiber, the contact angle experiment designed in this paper refers to the previous work [26]. It is generally accepted that the YLCA value of θ < 90° represents the surface having an affinity towards the liquid, so the surface is normally called hydrophilic. A value of θ_w_ > 90° shows the nonwetting characteristic of the surface, or the surface showing less affinity towards the liquid; therefore, such surfaces are called hydrophobic [27]. Furthermore, evidence is given in this paper that the apparent contact angle is influenced by increasing the roughness on the surface of fiber at same time [28]. When the water droplet contacts the PPTA fiber, it will form a similar clamshell conformation at the contact site, which is obviously different from that of the TLCP fiber. This is very similar to the Wilhelmy plate method to measure YLCA [29]. During that measurement, the partially immersed membrane is pulled out of the liquid vertically. At this time, the film and the liquid will form a similar kind of adhesion; therefore, referring to the contact angle defined in the Wilhelmy plate method, we will use the contact angle shown in Figure 2b.

Through the experiments designed in Section 2.3.4, it is observed that the water droplets are obviously adhered to the edge of the PPTA fiber. The software attached to the instrument has been utilized to measure the YLCA between the pristine PPTA fiber and the droplet of water, and the value is 40°; similarly, the YLCA attributed to the pristine TLCP fiber is 125°. Although there is no difference from the profile of the droplet, there is adhesion between the PPTA fiber and the droplet at the position indicated by the dashed red circle in Figure 3, while the TLCP fiber does not exhibit this phenomenon. It can be preliminarily judged that the chemical structure of the PPTA fiber should produce hydrophilicity greater than that of the TLCP fiber.

According to Section 2.3.5, the moisture on the surface of the two kinds of fibers is dried firstly by high-speed centrifugation. Table 2 lists the moisture content (MC) after each dehydration. After eight rounds of dehydration, the MC of the PPTA fibers decreases from 13.67% to 4.83%, and gradually becomes close to stability. The MC of the TLCP fibers decreases from 5.82% to 0.2% after seven rounds of dehydration, and becomes stable. From the comparison in Figure 4, it can be considered that under the same conditions, the difference in the MC of the two kinds of fibers reaches an order of magnitude, so it can be concluded that more moisture can be retained in the PPTA fibers, whose moisture is much greater than that of the TLCP fibers.

These experiments not only further prove that the hydrophilicity of PPTA fibers surpasses the opponent in another aspect, but also that PPTA fibers are superior to TLCP fibers in terms of the water absorption for this reason. This is confirmed by the evidence in Figure 5. Here, we use yellow lines represent microfibrils. The water molecules and the amide bonds in the microfibrillar structure of the PPTA fiber are tightly connected due to the action of hydrogen bonds. Although the contact between the droplet and the monofilament is only micron-scale adhesion, it is unexpected that the final water absorption rate is significantly different. Therefore, it is foreseeable that the cable made of PPTA fiber will absorb more water in the marine operating environment.

### 3.2. Analysis of the Failure of Liquid Crystal Fibers’ Mechanical Properties

Duo to the natural conditions, the average winter temperature in the Arctic circle is −30 °C, but it is not quite as cold as the average winter temperature in the Antarctic circle, which hovers around −60 °C. The ocean surface waters are generally very cold in proximity to the Arctic or Antarctic circle, and close to freezing. However, pockets of substantial heat often lurk at depth, preserved by higher salinity that makes water warmer; these areas have a temperature ranging from −2 to 4 °C. The ropes of the mooring system operating in this environment must experience the following two situations: first, frequent exposure into seawater and air, so that the immersed cables are frozen and then melted by the sea water; second, the frictional heat caused by the high-speed winding from the winch, which makes the ice slag in the frozen part melt more quickly. Taking the operating temperature range of −60–4 °C and the high frequency into consideration, first of all, we design three freezing conditions and melting media to match the temperature range of the ropes used in cold marine operations. Use of liquid nitrogen and hot water is considered only for more extreme conditions. Under each freezing condition, we used the time of different freeze–thaw (F–T) cycles to simulate the rapid thawing caused by frictional heating. Because of the high frequency of operations, we try to increase the number of F–T cycles as much as possible, up to a limit of 700 times.

The retention of tensile strength and the initial modulus for the two sorts of fibers are shown in Figure 6. First of all, after seven experiments, the mechanical properties of TLCP fibers show almost no loss. Except for experiment No. 1, the mechanical properties of the PPTA fibers after immersion in hot water for 24 h show no loss as well; however, their mechanical properties undergo a decline during the subsequent six experiments. The freezing media of experiments No. 2 and 3 are both refrigerators. The thawing conditions are designed so that the fibers are immersed in room temperature water, and the F–T temperature span is 20 °C; an F–T cycle lasts 40 min and is repeated 15 and 50 times, respectively. The retention of tensile strength is around 85%, while the retention of the initial modulus is 93% and 91%, respectively.

The freezing medium in Experiment No. 4 is dry ice. The thawing conditions are to immerse the fibers in room temperature water, with an F–T temperature span of 65 °C and an F–T cycle of 5 min, repeated 50 times. The retention of tensile strength and initial modulus is around 81% and 86%, respectively.

The freezing media in experiments No. 5, 6 and 7 are liquid nitrogen. The thawing conditions are designed so that the fibers are immersed in room temperature water and hot water, respectively. The freezing and thawing temperature span is 215, 250, and 290 °C, and F–T cycles of 10 and 5 s were repeated 110, 30 and 700 times, respectively. The result show that the retention rate of the tensile strength of PPTA fibers is around 76%, and the retention rate of the initial modulus is around 82%.

In fact, there have been many publications on the performance of high-performance liquid crystal fibers, mainly covering heat resistance, chemical resistance and photochemical erosion and modification, such as the preparation of high-temperature chemical industry filter materials [30]. Due to the chemical inertness of the Kevlar^®^ fiber surface, so that the interface adhesion between the fiber and the matrix resin is poor, fiber surface modification is carried out to improve the mechanical properties of the composite material [31]; Kelvar^®^ and Vectran^®^ fiber are in great demand in the aerospace field, so the modification of their UV resistance has been widely studied by scholars [32,33].

However, we show that the most common water and ice (through a series of temperature ranges, F–T frequencies and other combinations of parameters to design experiments) can also cause the loss of mechanical properties of powerful liquid crystal fibers such as PPTA fiber; this does not feature in previous research. There are many research directions that have been ignored by scholars.

From the results of the above experiments, it can be seen that, according to the freezing medium, that is, the minimum freezing temperature (−15, −50 and −200 °C), the mechanical properties of PPTA fibers are also decreased down three steps, that is, the retention of tensile strength drops to around 85%, 80% and 75% of the original, respectively. Considering that the condition of No. 7 is almost exaggerated to the extremity, and the results are basically similar to the No. 5 and No. 6, we think that after the experiment of No. 7, even if the F–T effect is intensified, the failure of mechanical properties belonging to the PPTA fiber will remain stable.

We intend to survey the data of the degree of crystal orientation and the FT-IR spectra about the two kinds of fibers to see if we can reach some conclusions.

### 3.3. The Degree of Crystal Orientation of Two Types of Fibers after F–T Experiments

To determine whether the orientation state changed after the F–T tests, we carried out 2D-WXRD characterization of the fibers. According to the data tested by researchers many years ago, the degree of crystal orientation of Vectran^®^ fiber through utilizing 2D-WAXD is about 96% [34]; recently, some scholars have prepared TLCP fiber (Seyang Polymer Co., Ltd., Incheon, Korea) through different melt spinning conditions. The results are about 90–95% [35]; after different stretching conditions, the degree of crystal orientation measured in Kelvar^®^ Fiber is about 80–90%, when using the same method [36]. After these F–T experiments, representative samples, i.e., No. 3, No. 4 and No. 7, are selected, and the peak intensities of the equatorial lines (001) of the two kinds of fiber measured by 2D-WAXD are shown in Figure 7. The comparison of the degree of crystal orientation of the two kinds of fiber, calculated according to Equation (1), has already been shown in Figure 8; the specific values are shown in the following Table 3. According to the degree of crystal orientation measured by 2D-WAXD, it can be seen that the data range is in line with the previous research results. The values of TLCP fibers and PPTA fibers are around 92% and 89%, respectively, and these results show that the value of the orientation degree has almost no fluctuation; we can infer that the orientation state structure of the fibers did not change after the F–T experiments.

### 3.4. The ATR-FTIR Analysis of TLCP and PPTA Fibers after the F–T Experiments

To verify whether the chemical structure of the fibers changed after F–T experiments, we performed ATR FT-IR characterization on the fibers. Since TLCP fiber belongs to “wholly aromatic polyesters”, the FTIR spectrum shows some peaks related to ester functional groups in the benzene ring, C–H and C–O bonds. The peaks located at 1600–1850 cm^−1^ are related to carbonyl groups, while the peaks located at 1200–1700 cm^−1^ are related to the skeleton stretching vibration of the aromatic rings. As shown, the FTIR spectrum of TLCP fiber includes typical absorption peaks at 1732 cm^−1^ (C=O stretching vibration) and 1053 cm^−1^ (C–O stretching vibration). The absorptions at 1632 and 1473 cm^−1^ are due to the C=C stretching vibration of the HNA unit in the copolymer chain, while the peaks at 1601, 1506 and 1414 cm^−1^ correspond to the C=C stretching vibration of the HBA unit in the TLCP fiber [37]. The peaks at 885 and 756 cm^−1^ are due to the characteristic C-H deformation vibration in the aromatic ring, and the strong absorption peaks at 1259, 1182, 1157 and 1014 cm^−1^ are due to the C–O–C asymmetric stretching vibration [38].

The FT-IR spectrum of Kevlar^®^ fibers allows the following band assignments: 3320 cm^−1^, the N-H stretching vibration in trans-amide with hydrogen bonds; 3054 cm^−1^, the C–H stretching vibration in unsaturated compounds; 1646 cm^−1^, the C=O stretching vibrations for amide groups forming hydrogen bonds (commonly referred to as amide I band); 1543 cm^−1^, the coupled modes of N–H deformation and C–N stretching vibrations; 1608 and 1515 cm^−1^, the C=C stretching vibration of aromatic rings; 1018 cm^−1^, the in-plane C–H vibrations of aromatic rings (which are characteristic of para-substituted aromatics, especially polyamides); 827 cm^−1^, the C–H vibrations of two adjacent out-of-plane hydrogens in aromatic rings; and 526, 729 and 865 cm^−1^, out-of-plane N–H deformation vibration. One absorption band at 3150 cm^−1^ can be attributed to the N–H vibration of cis-amide; the absorption band at 1317 cm^−1^ can be attributed to Ph-N vibration [39].

After the F–T experiments, the FT-IR spectra of the two kinds of fiber measured by ATR FT-IR are shown in Figure 9. TLCP fibers select the relatively stable benzene ring skeleton group at about 1601 cm^−1^ as the reference peak, while PPTA fibers select the vibrational band of the aromatic group at 1515 cm^−1^ as the reference band. By comparing the normalized absorbance intensity [40], the peak shape and peak intensity of the two kinds of fiber did not change under different experimental conditions, and no new absorption peak appeared, indicating that the PPTA fibers are not affected by the hydrolysis reaction [16]; additionally, of course, there is no change in the chemical structure of the TLCP fibers. It seems that the chemical structure and the orientation structure of the crystal region of the two kinds of fibers have not changed after repeated extreme F–T experiments. We need to design further experiments to unravel the mystery of their loss of mechanical properties.

### 3.5. The Observation and Analysis of SEM Measurement Results

As shown in Figure 10a,b, the surface of the two kinds of fiber after being immersed in hot water is still as smooth as before. Then, in Figure 10c–f, the surface of the PPTA fibers frozen by the refrigerator has a slight roughness, while the TLCP fibers are still relatively smooth. The surface roughness of the PPTA fibers frozen by dry ice increased, and microcracks appeared; in contrast, the appearance of TLCP fiber did not change too much. All of these results are shown in Figure 10g,h. It is found that with the aggravation of freezing and thawing conditions, after freezing with liquid nitrogen, the damage on the surface of the PPTA fiber increases, and deep microcracks and ravines appear; however, the damage to the TLCP fiber is quite minimal. We observed these phenomena in Figure 10i–n.

We can observe that as the temperature decreases, the two kinds of fiber undergo freezing by a refrigerator, dry ice and liquid nitrogen, and the freezing temperature gradually drops from −15 °C, −50 °C to −200 °C, causing obvious damage to the surface of PPTA fiber. Gullies are formed, and the TLCP fibers have little influence. The level of surface damage is also similar to the three stages of the previous decrease in the mechanical properties of the PPTA fibers.

Furthermore, in previous works, it has been proven that as the roughness of the fiber fabric surface increases, its hydrophilicity will be greatly improved [28] as well. Subsequently, the YLCA assigned to PPTA fiber and TLCP fiber after the F–T experiments is 35° and 125°, as indicated by the dashed red circles, respectively, which are shown in Figure 11. This shows that the microcracks on the surface of the PPTA fiber cause roughness and increase the hydrophilic effect, which is also the reason for the further decline in the YLCA.

Therefore, it can be considered that the lower the freezing temperature, the greater the surface damage to the PPTA fibers after F–T experiments. When the surface of the PPTA fiber deteriorates, its mechanical properties inevitably decline.

### 3.6. Discussion of EDS Elemental Analysis

From the above data of YLCA, it has been proven that the PPTA fiber is more hydrophilic; after the F–T experiments, the surface morphology is destroyed, resulting in greater water absorption. Considering the microfibrillar structure of the liquid crystal fiber, we estimate that the moisture could penetrate into the interior of the PPTA fibers.

In the early years, when the scholar Sawyer studied liquid crystal fibers, he showed the “fibril” microstructure oriented along the drawing direction, observed by scanning electron microscope (SEM) [41]. With advancements in higher spatial resolution imaging techniques, Sawyer used TEM (transmission electron microscopy) and STM (scanning tunneling microscopy) to determine that Kevlar^®^ and Vectran^®^ fiber consist of long ribbon-like filaments with hierarchy “microfibril” structures, typically 22 nm wide and 3–5 nm thick [42]. The biggest feature of this type of liquid crystal fiber is the rigid-rod like chemical structure of the polymer molecule, which produces the morphology of the “microfibril”. Compared with the flexible chain polymer, there is almost no entanglement. This is the one of the reasons that the structure of the microfibrils may be destroyed due to the expansion in volume after water penetrates and freezes.

At room temperature, water wraps liquid crystal fibers with a microfibrillar structure, as shown in Figure 12a. Here, we use yellow lines represent microfibrils, and gray bundles represent larger microfibers as simulations. When water is in a liquid state at 25 °C, the average number of hydrogen bonds in a water molecule is 3–4 bonds. The hydrophilic properties of PPTA fibers make them more closely bound with water molecules through hydrogen bonds, so water molecules can penetrate in the microfibrillar structure of this liquid crystal fiber. In Figure 12b, during the freezing process, the mobility of water molecules is reduced, allowing more stable hydrogen bonds to form between them. Since the water molecules are affected by the lattice structure once the freezing point is reached, the average number of hydrogen bonds per water molecule becomes the maximum value, namely, 4 bonds; this ordered crystallization leads to a decrease in density, as the distance of each water molecule from its neighbors is equal to the length of the hydrogen bonds. Therefore, water expands in volume when it freezes. This will gradually destroy the unique microfibrillar structure of PPTA fiber and cause roughness and cracks in the surface. After frequent F–T experiments, more and more water molecules will penetrate into the larger and larger pores caused by the expansion of the water frozen volume, and a vicious circle will damage the microfibrillar structure of the PPTA fibers step by step, resulting in the decline in their mechanical properties, as shown in Figure 12c,d.

The damage on the surface can be observed by SEM; however, when the fiber undergoes F–T experiments, larger and deeper damage may be incurred inside. We intend to use water-soluble cations to penetrate into these microcracks, and then perform quantitative characterization to reveal the damage caused by the F–T experiments. The method is to soak the two kinds of fiber after the F–T experiment (we chose the sample of No. 3 as it is the lightest F–T cycle condition) with the same concentration of sodium chloride solution, instead of water, for 24 h. After washing with the same deionized water and drying, half of the surface layer of the fiber is peeled off along the fiber axis under the microscope. Finally, the content of sodium ions in the surface layer and the subsurface of the PPTA and TLCP fiber are tested by EDS elemental analysis.

The positions of EDS observation and the mass percentages content of sodium ions for PPTA fibers and TLCP fibers are shown in Figure 13. According to the content of sodium ions at three different locations on the fiber surface, the values (from yarn center to its edge) on the PPTA fiber are 1.02%, 1.09%, and 1.83%, respectively, which are higher than the values on the same sites belonging to the TLCP fiber: 0, 0.1% and 0.3%. For the content of sodium ions at three different positions in the fiber sublayer, the values (from yarn center to its edge) for PPTA fiber are 0.59%, 0.69% and 0.42%, respectively, while for TLCP fiber they are 0, 0.17% and 0.01%, i.e., much lower. A comparison of the sodium ion contents of the two fibers at the same position is shown in Figure 14.

Therefore, it can not only be proven that the surface of PPTA fiber finds it easier to absorb water molecules; the sodium ions have indeed penetrated into the interior of the microfibrillar structure by means of water. The content of sodium ions belonging to the PPTA fiber in six different positions is much higher than the same positions of the TLCP fiber, indicating that in the previous test of moisture content, the amount of water infiltrated into the PPTA fibers is higher than that in the TLCP fibers. This is the result of the frequent volume changes during the freezing process of water after intense and repeated F–T experiments; thus, the destruction of the microfibrillar structure of the liquid crystal fibers and the loss of mechanical properties are incurred.

## 4. Conclusions

In this study, after F–T experiments simulating extreme environments, the mechanical properties of the PPTA fibers decreased by up to 25%, while the F–T experiments had little effect on the TLCP fibers. The chemical and orientation structures of the crystal region belonging to these fibers were not damaged according to the FT-IR analysis and data from the degree of crystal orientation. Under SEM observation, it was found that the surface of PPTA fibers has greater roughness and even cracks than TLCP fibers; PPTA fibers are more hydrophilic than TLCP fibers due to the chemical structure of the polymer molecules, resulting in better water absorption. Therefore, the volume expansion of ice after the process of F–T experiments will damage the surface of PPTA fibers, and even allow water molecules to penetrate further into the subsurface, resulting in the destruction of the unique ribbon-like microfibrillar structure belonging to liquid crystal fibers. For the two kinds of fibers immersed in the same concentration of sodium chloride solution, it is found that the content of sodium ions on the surface and subsurface of the PPTA fiber was much higher than that of the TLCP fiber after utilizing the EDS elemental analysis, which proved that the water absorption of the PPTA fibers was further increased. Based on all the above results, this should eventually lead to the loss of the mechanical properties of PPTA fibers. Therefore, in high-latitude marine operations, TLCP fibers may be more competent.

## Figures and Tables

**Figure 1 polymers-15-02001-f001:**
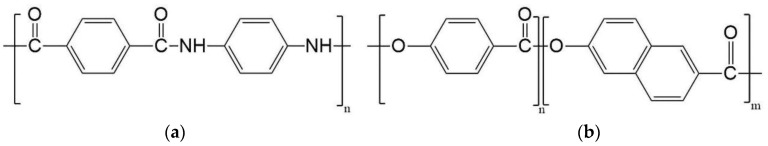
The chemical structure of PPTA fiber (**a**) and TLCP fiber (**b**).

**Figure 2 polymers-15-02001-f002:**
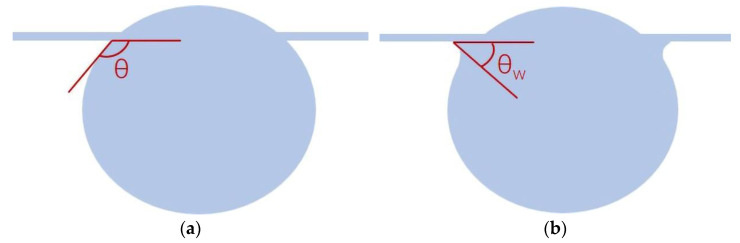
The YLCA between the yarn and droplet: (**a**) not wetted, (**b**) wetted.

**Figure 3 polymers-15-02001-f003:**
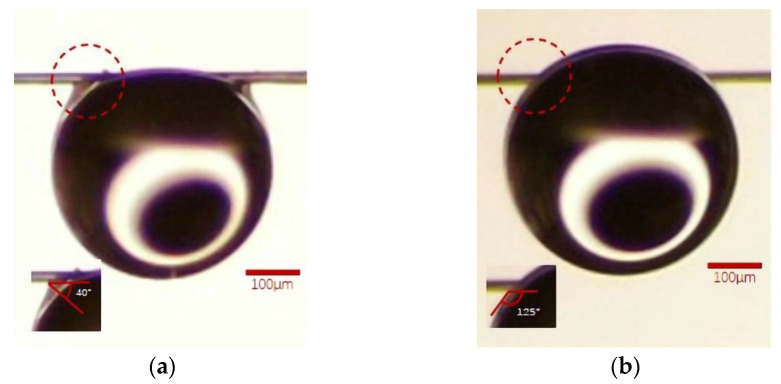
The YLCA of two kinds of pristine fiber: (**a**) PPTA fiber, (**b**) TLCP fiber.

**Figure 4 polymers-15-02001-f004:**
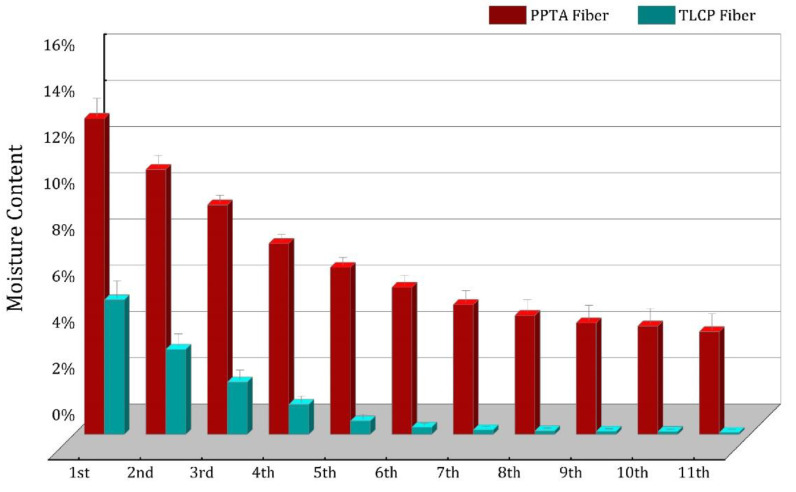
Comparison of moisture content of two fibers after centrifugal dehydration.

**Figure 5 polymers-15-02001-f005:**
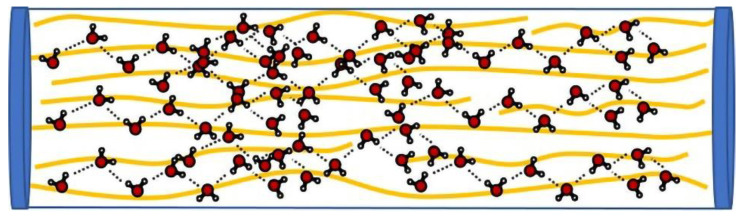
The water molecules connected to microfibrillar structure of the PPTA fiber by the action of hydrogen bonds.

**Figure 6 polymers-15-02001-f006:**
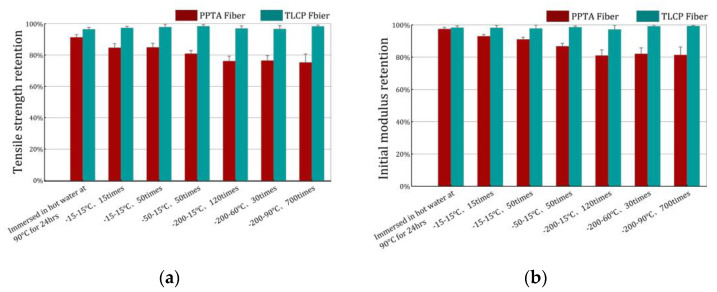
The retention of tensile strength (**a**) and initial modulus (**b**).

**Figure 7 polymers-15-02001-f007:**
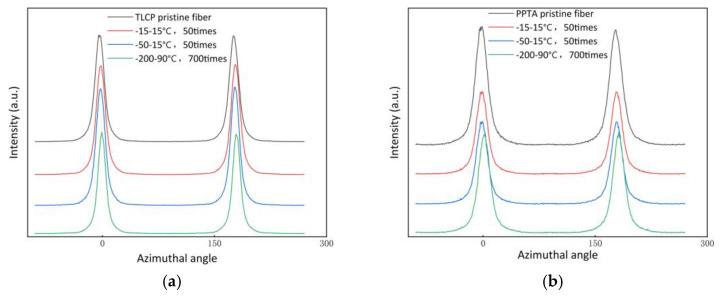
The peak intensities of the equatorial lines belonging to TLCP fibers (**a**) and PPTA fibers (**b**).

**Figure 8 polymers-15-02001-f008:**
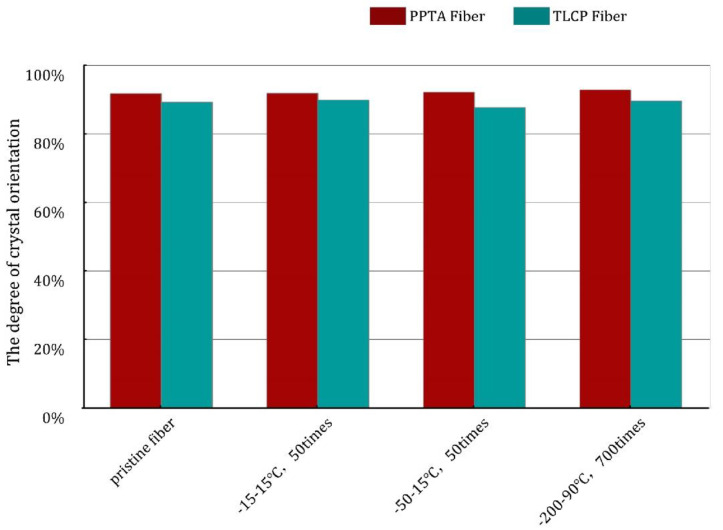
The comparison of the degree of crystal orientation in liquid crystal fibers.

**Figure 9 polymers-15-02001-f009:**
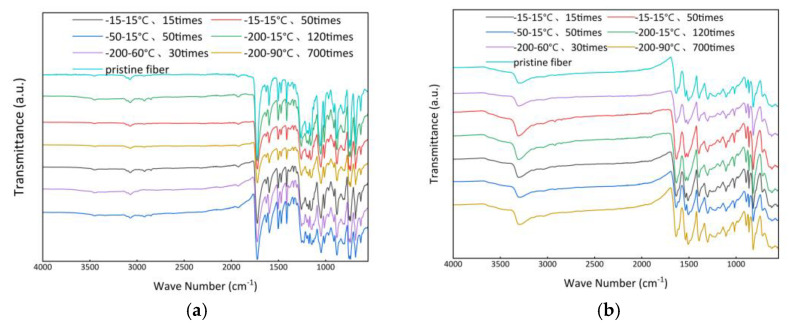
FT-IR spectra after the F–T experiments: (**a**) TLCP fibers, (**b**) PPTA fibers.

**Figure 10 polymers-15-02001-f010:**
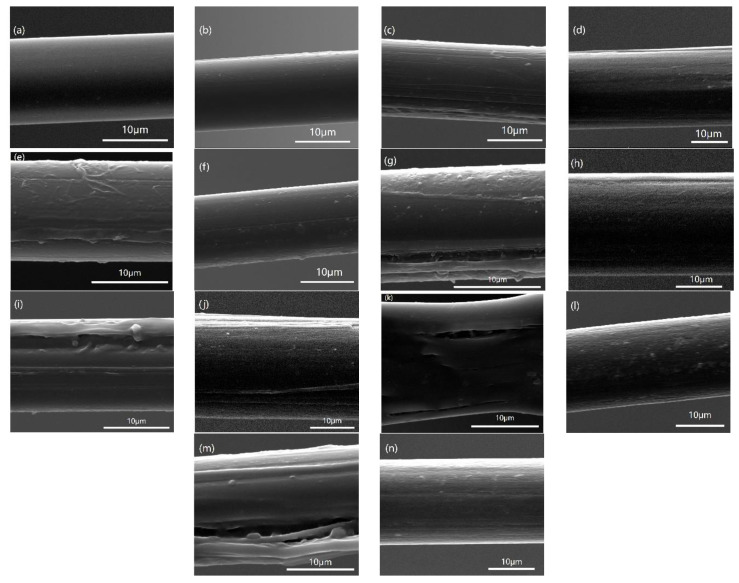
SEM microstructures of PPTA fibers and TCLP fibers with F–T experiment 1: (**a**,**b**); F–T experiment 2: (**c**,**d**); F–T experiment 3: (**e**,**f**); F–T experiment 4: (**g**,**h**); F–T experiment 5: (**i**,**j**); F–T experiment 6: (**k**,**l**); F–T experiment 7: (**m**,**n**), respectively.

**Figure 11 polymers-15-02001-f011:**
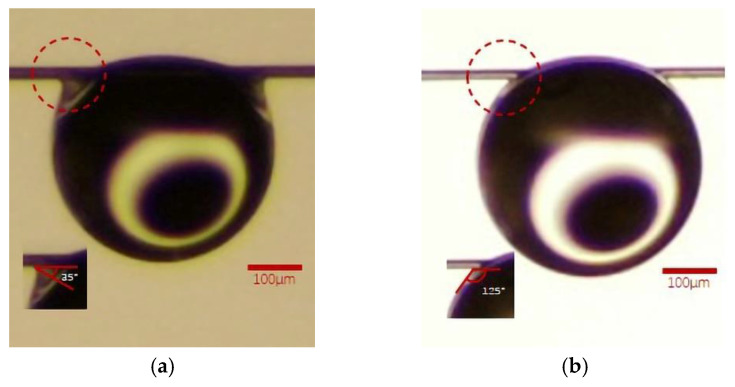
The YLCA of two kinds of fiber after the F–T experiments (**a**) PPTA fiber, (**b**) TLCP fiber.

**Figure 12 polymers-15-02001-f012:**
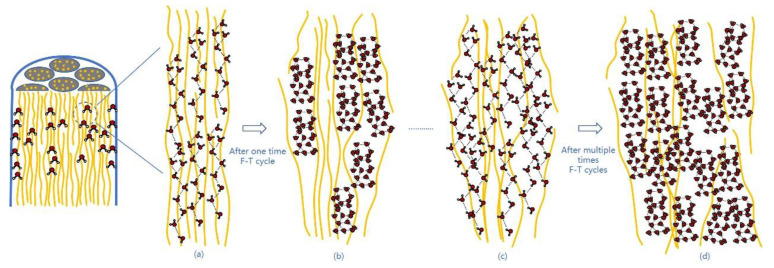
The scheme of PPTA fiber under F–T experiments: (**a**) water molecules are linked to the microfibrillar structure of PPTA fiber through hydrogen bonds after the PPTA fiber is immersed in water; (**b**) the volume expansion of ice destroys the microfibrillar structure after one F–T cycle; (**c**) more water molecules penetrating into the microfibrillar structure of PPTA fiber; (**d**) larger microcracks caused after multiple F–T cycles.

**Figure 13 polymers-15-02001-f013:**
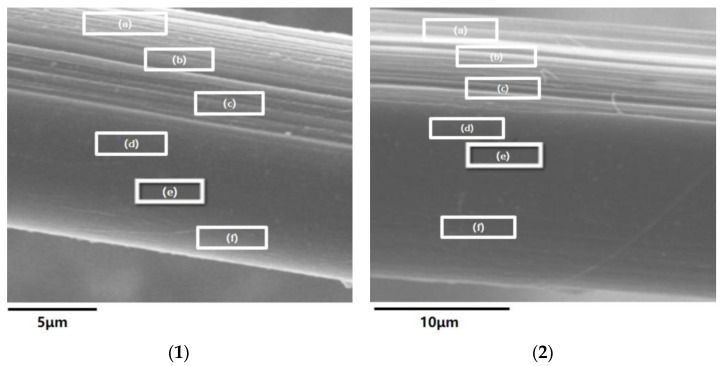
The different sites of EDS elemental analysis, from the top to the bottom in the figure are: (a)–(f); PPTA fiber belongs to (**1**) and TLCP fiber belongs to (**2**).

**Figure 14 polymers-15-02001-f014:**
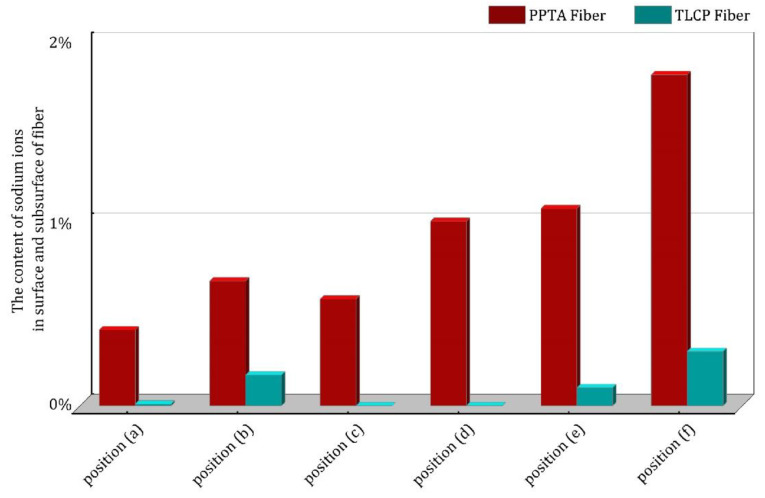
The content of sodium ions on the surface and subsurface of the two kinds of fibers.

**Table 1 polymers-15-02001-t001:** Sample codes of TLCP and PPTA fiber in the F–T experiments.

Sample Code	Experiment Condition (°C)	Thawing Medium	F–T Cycle Times	Temperature Range (°C)	One F–T Cycle Time Duration
No. 1	Hot water (90)	-	-		24 h
No. 2	Refrigerator (−5)	Room temperature water (15 °C)	15	20 °C	40 min
No. 3	Refrigerator (−5)	Room temperature water (15 °C)	50	20 °C	40 min
No. 4	Dry ice (−50)	Room temperature water (15 °C)	50	65 °C	5 min
No. 5	Liquid nitrogen (−200)	Room temperature water (15 °C)	110	215 °C	10 s
No. 6	Liquid nitrogen (−200)	Hot water (60 °C)	30	250 °C	10 s
No. 7	Liquid nitrogen (−200)	Hot water (90 °C)	700	290 °C	5 s

**Table 2 polymers-15-02001-t002:** Moisture content data of two fibers after centrifugal dehydration.

	PPTA Fiber	TLCP Fiber
1st	13.67%	5.82%
2nd	11.47%	3.68%
3rd	9.93%	2.26%
4th	8.26%	1.30%
5th	7.23%	0.58%
6th	6.36%	0.31%
7th	5.62%	0.20%
8th	5.15%	0.15%
9th	4.83%	0.14%
10th	4.69%	0.12%
11th	4.45%	0.09%

**Table 3 polymers-15-02001-t003:** The degree of crystal orientation in the two fibers, as measured by 2D-WAXD.

Sample	TLCP Fiber	PPTA Fiber
Pristine fiber	91.8%	89.3%
−15–15 °C, 50 times	91.9%	89.9%
−50–15 °C, 50 times	92.2%	87.7%
−200–90 °C, 700 times	92.9%	89.6%

## Data Availability

Not applicable.

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
