# Peer review of "Implication of Freeze–Thaw Erosion and Mechanism Analysis of High-Performance Aromatic Liquid Crystal Fibers"

_polymers, 2023, doi:10.3390/polym15092001_

Round 1
Reviewer 1 Report
Before submitting the revised version of this work, the following items must be addressed:
1. The authors' efforts to clearly mention more details about the prepared fibers are greatly appreciated; is this the first time these types of fibers have been prepared using this process?
2. It is important that the originality of this work be brought out in the open throughout the improved text.
3. The authors are appreciative that they have the possibility to enhance the quality of all of the images, despite the fact that the quality of some of the figures (Figs. 3, 6, 8, and 11) cannot be clearly observed. Resolution of at least 600 DPI or greater is greatly appreciated.
4. During the experimental stage, it is important to consider the degree of purity of all of the chemicals and reagents that are utilized. In addition to that, the molecular weight of the polymers that were utilized can be included as well.
5. It is possible for the authors to include it within the discussion section as an illustrative scheme.
6. The work has excellent writing throughout. However, there are a few problems that were discovered in the text, and the authors are respectfully requested to reread the entire amended text for any typos or grammatical errors that may have been missed.
7. The graphical abstract is very valuable for this task since it will provide the reader with a solid understanding of the information contained inside the document. Because of this, the authors deserve a lot of praise for coming up with a graphic abstract that is original and intriguing for this particular piece of work.
Author Response
Dear Editor and Reviewers,
Thank you for your suggestion for our manuscript (polymers-2255387). And we have amended our manuscript and responded as follows. The change in the manuscript was made in RED (in the Revised manuscript with changes marked).
Response to the Reviewer #1’s Comments point by point:
------------------------------------------
Reviewer’s comment 1:
- The authors' efforts to clearly mention more details about the prepared fibers are greatly appreciated; is this the first time these types of fibers have been prepared using this process?
Response:
Thanks for your comment. The two high-performance liquid crystal fibers involved in the article are prepared by Chinese enterprises: TLCP fibers which are provided by Ningbo Higlar New Material Technology Co.,Ltd (China) have started production in 2019; PPTA fibers which provided by Yantai Tayho Advanced materials Co.,Ltd (China) have put into production in 2011. TLCP fibers and PPTA fibers were prepared by melt spinning and solution spinning respectively. the tensile strength and initial modulus of the PPTA and TLCP fiber are: 17.47cN/dtex, 545.84 cN/dtex; 21.34 cN/dtex, 722.86 cN/dtex, respectively.
Reviewer’s comment 2:
- It is important that the originality of this work be brought out in the open throughout the improved text.
Response:
First of all, thank you very much for your reminder. The originality of this article is special (from r.132 to r.138): Compared with the heat resistance, chemical and UV resistance research of high-performance liquid crystal fibers, there is no literature to provide sufficient data to prove that, as representative products of high-performance fiber——PPTA and TLCP Fiber in frequent freeze-thaw cycles, the stability of its mechanical properties. It is possible that this is the first study of the effect of ordinary water and temperature changes on the mechanical properties of high-performance liquid crystal fibers. Therefore, I have added the above text to the original paragraph.
Reviewer’s comment3:
- The authors are appreciative that they have the possibility to enhance the quality of all of the images, despite the fact that the quality of some of the figures (Figs. 3, 6, 8, and 11) cannot be clearly observed. at least 600 DPI or greater is greatly appreciated.
Response:
Indeed, I overlooked something before, thanks again for your suggestion. Resolution of all these figures have been changed to 600DPI.
Reviewer’s comment 4:
During the experimental stage, it is important to consider the degree of purity of all of the chemicals and reagents that are utilized. In addition to that, the molecular weight of the polymers that were utilized can be included as well.
Response:
Thanks for your suggestion. And we have added the purity information about the chemicals and reagents that we used in the revised manuscript. In 2.1 Materials section, I have added "Sodium chloride (AR≥99.5%) is obtained from Shanghai Lingfeng Chemical reagent Co.,LTD.(China). It is then made into a 10% solution with deionized water." For TLCP fiber, Ubbelohde viscometer is usually used to measure its viscosity-average molecular weight, and highly toxic and highly polar pentafluorophenol is used as a solvent. However, the TLCP fiber used in this article was heat-treated, and it was more difficult to dissolve in this solvent, so basically no convincing data could be obtained. However, we will study better solvents to improve this work in the future.
Reviewer’s comment 5:
It is possible for the authors to include it within the discussion section as an illustrative scheme.
Response:
Thanks for your advice. A lot of content is described in the experimental part, which makes the key points of the article insufficient. I have made the following modifications:
1,The first paragraph of section 2.2 has been moved to the first paragraph of section 3.2.
2,The explanatory text of section 2.3.4 has been moved to section 3.1.
All these changes were marked in red.
Reviewer’s comment 6:
The work has excellent writing throughout. However, there are a few problems that were discovered in the text, and the authors are respectfully requested to reread the entire amended text for any typos or grammatical errors that may have been missed.
Response:
Thank you for your reminding, I have carefully checked and amended the manuscript.
Reviewer’s comment 7:
The graphical abstract is very valuable for this task since it will provide the reader with a solid understanding of the information contained inside the document. Because of this, the authors deserve a lot of praise for coming up with a graphic abstract that is original and intriguing for this particular piece of work.
Response:
Thank you for your reminding. The graphical abstract has been produced as shown in the following figure:

Reviewer 2 Report
The study is really comprehensive and I can say this paper is among the few papers available in the area. Congratulations on that. As your work is relevant to the Antarctic conditions in some way and all the experimental work designed base don this, you can also carry out Antarctic clothing assessment that is produced or integrated with crystal fibers and it could be of a great interest. I can advice you one paper for your future studies into this area:
Sivri, Ç., Gül, S., & Aksu, O.R. (2022). A Novel Pythagorean Fuzzy Extension of DEMATEL and Its Usage on Overcoat Selection Attributes for Antarctic Clothing. Int. J. Inf. Technol. Decis. Mak., 21, 821-850. can be reached from the following link: https://www.worldscientific.com/doi/10.1142/S021962202250002X
Author Response
Dear Editor and Reviewers,
Thank you for your suggestion for our manuscript (polymers-2255387). And we have amended our manuscript and responded as follows. The change in the manuscript was made in RED (in the Revised manuscript with changes marked).
Response to the Reviewer #2’s Comments point by point:
------------------------------------------
Reviewer’s comment 1:
The study is really comprehensive and I can say this paper is among the few papers available in the area. Congratulations on that. As your work is relevant to the Antarctic conditions in some way and all the experimental work designed base don this, you can also carry out Antarctic clothing assessment that is produced or integrated with crystal fibers and it could be of a great interest. I can advice you one paper for your future studies into this area:
Sivri, Ç., Gül, S., & Aksu, O.R. (2022). A Novel Pythagorean Fuzzy Extension of DEMATEL and Its Usage on Overcoat Selection Attributes for Antarctic Clothing. Int. J. Inf. Technol. Decis. Mak., 21, 821-850. can be reached from the following link: https://www.worldscientific.com/doi/10.1142/S021962202250002X
Response:
Thank you for the materials you provided, which are very useful for my future research. I have included the article (from r.79 to r.82) as a reference as follows:
Clothes for scientific exploration in the Antarctic region have recently been studied1. These clothes are expected to withstand harsh conditions, and they must have certain functions to ensure the safety and comfort of scientists. Actually, this study inspired me.
- Sivri Ç, Gül S and Aksu OR. A novel Pythagorean fuzzy extension of DEMATEL and its usage on overcoat selection attributes for Antarctic clothing. International Journal of Information Technology & Decision Making 2022; 21: 821-850.

Reviewer 3 Report
Comments on the manuscript (Polymers-2255387)
Manuscript (Polymers-2255387) reports the implication of freeze-thaw erosion and its mechanism of fully aromatic polyamide (PPTA) and polyester (TLCP) liquid crystal fibers. In F-T cycles under different conditions, PPTA fiber was damaged and lost its physical performance more than TLCP fiber. Due to their inherent chemical structure, PPTA fibers are more hydrophilic than TLCP fibers, so PPTA fiber absorb a higher amount of water. The proposed mechanism is based on the volume expansion of water at the phase transition to ice during F-T experiments.
The subject matter is interesting and experiments were performed well. The manuscript is well organized and the main concepts are supported by the experimental data. Therefore, it is recommended that this paper be published in the MDPI Polymer. However, the manuscript should to be improved in a number of ways, and the authors are encouraged to revise the manuscript as suggested in the following comments.
Minor comments:
1. Abbreviations must be defined the first time they appear. For examples, LLC on line 52 and PA6 on line86 should be defined.
2. What was the diameter and length of the fibers used in the study? It would be nice if these information were provided in Materials section on page 4.
3. Is it “In Korea” or “in Korea” on line349?
4. Is the word order of “partial the reason” on line 451 correct?
5. On line 457, what is “liquid fiber”? Is it “liquid crystal fiber”?
6. Please, check “date” on line 513. It seems to be “data”.
Major concerns:
I believe the experiment worked well. Overall, authors proposed reasonable explanations for the results. However, there are concerns about the length scale and clarity of the proposed model.
1. First of all, the length scale is not clear in the presentation. For readers’ clear understanding, it will be better to provide the diameter of fiber and microfibrils, respectively, in Figure 5. The yellow lines in Figure 5 appear to be crystalline microfibrils. If a fiber consists of crystalline bundles (i.e. microfibrils), what is the d-spacing (diameter) between the fibrils?
2. Do the yellow lines in Figure 5 and Figure 12 represent the same thing? What are they?
3. In Figure 7 and the corresponding descriptions, the equatorial lines should be defined with respect to the fiber axis. Quantitative values should be given for 2q-angle (Deg.) in Figure 7. Also, the corresponding d-spacing have to be discussed in the text to help readers better understand the fiber structure and length scale.
4. In Figure 12, the cross-section of a fiber shows cylindrical bundles (gray-shaded circles in the horizontal cross-section) and yellow lines in vertical section. However, for the gray shaded circles, there is no boundary in the vertical section. What are they? And where were water molecules absorbed/adsorbed? It should be clearly illustrated. Of the gray-shaded circles with yellow lines in it, which one is microfibril described in the text? Is this Figure consistent with Figure 5?
It is not clear whether yellow lines are polymer chain or microfibrils. As far as I understand, yellow lines seem to be described as microfibrils in the text and Figure 5. If this is the case, what are those gray bundles? Or, do yellow lines represent individual polymer chain and gray bundles represent microfibrils? This should be clarified both in Figure 5/12 and text.
5. What is the “aggregate structure” on line 512? There can be two different aggregate structures: (1) aggregate of polymer chains to form a microfibril and (2) aggregate of microfibrils to form a fiber. It will be better if it is clarified. The sentence on lines 512~513 is confusing. How can surface morphology change and cracks form without changing the aggregate structure?
6. Regarding the line 519, what is “the unique ribbon-like microfibrillar structure” in Figure 5 and 12?
7. Authors wrote that “Water molecules can penetrate in the microfibrillar structure of this liquid fiber” (line 457). What does this mean? It is not clear whether water can penetrate and be absorbed into the interstitial regions between microfibrillar bundles or whether water molecules can penetrate the microfibrils into polymer chains. The results will be very different in these cases. This difference needs to be more clearly discussed and substantiated by experimental results.
All of the above concerns are related to the fiber structure and erosion model. I may have misunderstood what has been described. However, this means that other readers may also get it wrong. Therefore, the above points should be clearly expressed so as not to cause misunderstanding.
Author Response
Dear Editor and Reviewers,
Thank you for your suggestion for our manuscript (polymers-2255387). And we have amended our manuscript and responded as follows. The change in the manuscript was made in RED (in the Revised manuscript with changes marked).
Response to the Reviewer #3’s Comments point by point:
------------------------------------------
Reviewer’s comment 1:
Abbreviations must be defined the first time they appear. For examples, LLC on line 52 and PA6 on line86 should be defined.
Response:
Thanks for your carful correction. After careful inspection, "Celanese Acetate LLC" is a typo, the correct one should be "Hoechst Celanese";the full name of PA6, Polyamide 6, has also been amended in the revised manuscript.
Reviewer’s comment 2:
What was the diameter and length of the fibers used in the study? It would be nice if this information was provided in Materials section on page 4.
Response:
Thanks for your suggestions. As shown in Figure 10, the diameter of the fiber is about 20 μm. The yarns used in this article are shown in the figure below, with the PPTA fiber on the left and the other on the right. The weight of each roll is about 1kg, and the linear density of the fibers is 1000dtex, so their total length is about 10000 meters, and all these information have provided in Materials section on page 4.

Reviewer’s comment 3:
Is it “In Korea” or “in Korea” on line349?
Response:
Thanks for correcting me again, it is definitely “in Korea”.
Reviewer’s comment 4:
Is the word order of “partial the reason” on line 451 correct?
Response:
After careful consideration, without changing the original meaning, I adjusted the sentence to “It is the one of the reasons that the structure of the microfibrils could be destroyed due to the expansion of the volume after the water penetrates and freezes.”.
Reviewer’s comment 5:
On line 457, what is “liquid fiber”? Is it “liquid crystal fiber”?
Response:
Definitely, it is liquid crystal fiber. I have corrected. Thanks for your help.
Reviewer’s comment 6:
Please, check “date” on line 513. It seems to be “data”.
Response:
This is an obvious typo, and I have corrected. I appreciate your meticulous help again.
Major concerns:
- First of all, the length scale is not clear in the presentation. For readers’ clear understanding, it will be better to provide the diameter of fiber and microfibrils, respectively, in Figure 5. The yellow lines in Figure 5 appear to be crystalline microfibrils. If a fiber consists of crystalline bundles (i.e., microfibrils), what is the d-spacing (diameter) between the fibrils?
Response:
Thank you very much for your suggestion. Figure 5 and Figure 12 are just a simulation presentation, not the real image of the electron microscope photo. In fact, the article did not measure the size of the specific microfiber, but relied on the support provided by the scholar Sawyer's previous classic research.
- C. Sawyer points out that LCP oriented extrudate structural hierarchy is composed of: macrofibrils, about 5μm diameter; fibrils, about 0.5μm (500nm) across and, microfibrils, that are about 0.05μm (50 nm) wide and 5 nm thick1, observed by scanning electron microscopy (SEM) and transmission electron microscopy (TEM). This hierarchy is illustrated in the LCP fiber structural model, shown in Fig.2(a).
About a decade later Sawyer went on to expand these ideas2: The microfibrils are tape-like in shape and this is hypothesized to be due to a replication of the rod-like molecular chain, and the specific model is shown in Fig. 2(b). Comparison of the molecular and microfibrillar sizes suggest that the microfibrils are composed of a minimum of two molecules, in the smallest dimension, and thus they represent the finest nanostructural element in the LCPs. An extended LCP structural model, consisting of well-ordered, elongated fibrils, continues to be consistent with measured properties: high anisotropy, very high tensile modulus and tensile strength, and poor shear and compressive properties in the lateral dimension.
Through Sawyer's research, it can be seen that microfiber is not a fixed size, but a hierarchical system. Similar to Figure 5 and Figure 12, the size of the microfibers represented by the simulation demonstration of the yellow line is gradually finer according to the resolution increase of different observation techniques.
|
|
|
Fig. 2 The layered, fibrous texture of the LCP material is shown in a model of the LCP structure (left) and a micrograph of the oriented state structure (right)1 |
|
|
|
Fig.3 An expanded structural model is shown for the thermotropic and lyotropic LCPs, with more detail of the microfibril sizes, shapes and order2 |
- Do the yellow lines in Figure 5 and Figure 12 represent the same thing? What are they?
Response:
As answered by Response1, the yellow lines in Figure 5 and Figure 12 of the article are the abstract embodiment of this hierarchy system. To be exact, they are the same simulation display used to represent the microfiber structure
- In Figure 7 and the corresponding descriptions, the equatorial lines should be defined with respect to the fiber axis. Quantitative values should be given for 2q-angle (Deg.) in Figure 7. Also, the corresponding d-spacing have to be discussed in the text to help readers better understand the fiber structure and length scale.
Response:
Thank you very much for your request, the Quantitative values of azimuthal angle have been added;
In fact, I did not analyze their crystallinity in the article for the following reasons:
- Liquid crystal polymers in the elongation flow field, that is, during spinning or extrusion, will lead to the formation of highly oriented elongated chain structures in the solid state3.
- Compared with the relatively short relaxation time of flexible chain polymer molecules, their disorientation time in the melt or solution is greatly increased4.
- Blundell and co-workers pointed out that the increase in tensile modulus of liquid crystal polymer materials is due to the increase in the level of macromolecular orientation, rather than from differences in chemical composition. The high degree of orientation of their rigid macromolecular chains is due to the extremely slow relaxation time, which maintains orientation to a greater extent than flexible chain polymers with faster relaxation times5.
Similarly, for the two liquid crystal fibers in this paper, their degree of orientation is also the key to determining the mechanical properties, rather than crystallinity. By the way, the estimated amount of crystallinity of TLCP at short times is about 8-10%6. The current shortcoming is that the orientation degree of the crystal region can only be provided by 2D-WAXD technology. That's why I didn't provide the d-spacing data related to crystallization, but you inspired me to further explore the size of the oriented microfibrils and the pores between them in future research.
- In Figure 12, the cross-section of a fiber shows cylindrical bundles (gray-shaded circles in the horizontal cross-section) and yellow lines in vertical section. However, for the gray shaded circles, there is no boundary in the vertical section. What are they? And where were water molecules absorbed/adsorbed? It should be clearly illustrated. Of the gray-shaded circles with yellow lines in it, which one is microfibril described in the text? Is this Figure consistent with Figure 5?
It is not clear whether yellow lines are polymer chain or microfibrils. As far as I understand, yellow lines seem to be described as microfibrils in the text and Figure 5. If this is the case, what are those gray bundles? Or, do yellow lines represent individual polymer chain and gray bundles represent microfibrils? This should be clarified both in Figure 5/12 and text.
Response:
Thanks for your reminding. The cylindrical bundles (gray-shaded circles in the horizontal cross-section) are also part of the abstract hierarchical system, but it is a relatively large part. Water molecules are adsorbed on the surface, and because of the voids in the microfiber structure, it penetrates into the interior of the fibers.
Explanatory text has been added in two places in the text.
- What is the “aggregate structure” on line 512? There can be two different aggregate structures: (1) aggregate of polymer chains to form a microfibril and (2) aggregate of microfibrils to form a fiber. It will be better if it is clarified. The sentence on lines 512~513 is confusing. How can surface morphology change and cracks form without changing the aggregate structure?
Response:
Your suggestion is very pertinent. The term aggregate structure is a bit ambiguous. In fact, what is characterized by 2D-WAXD technology is the orientation structure of the crystal region, which belongs to the tertiary structure in the main levels of polymer structure. Therefore, I changed “the aggregate structure” to “the orientation structure of the crystal region” in the revised manuscript.
At the beginning, I also thought that the damage on the surface would affect the orientation structure of the crystal region and further affect the mechanical properties, but in fact the result of the orientation structure did not change, which led to further exploration of the damage mechanism in this paper.
- Regarding the line 519, what is “the unique ribbon-like microfibrillar structure” in Figure 5 and 12?
Response:
"The unique ribbon-like microfibrillar structure" in Sawyer's paper, the expression of the microfibrillar structure is "tape-like", but in Yang Zhong's book7, he cites the same The article cited by Sawyer made this statement: ".....The ribbon-like texture of PLC extrudates was also described by Sawyer and Jaffe...". In addition, F. Sloan in his book8 also uses the expression: "...creating a rigid, ribbon-like...". Therefore, it is not a specific term and does not create ambiguity for the readers.
- Authors wrote that “Water molecules can penetrate in the microfibrillar structure of this liquid fiber” (line 457). What does this mean? It is not clear whether water can penetrate and be absorbed into the interstitial regions between microfibrillar bundles or whether water molecules can penetrate the microfibrils into polymer chains. The results will be very different in these cases. This difference needs to be more clearly discussed and substantiated by experimental results.
Response:
Thank you for your suggestion. Strictly speaking, the statement "Water molecules can penetrate in the microfibrillar structure of this liquid crystal fiber" is an inference drawn from the following three steps:
- The analysis of the hydrophilic and hydrophobic properties brought about by the different chemical structures of the two fibers;
- The moisture content of PPTA fibers is much higher than that of TLCP fibers (functional groups are ester groups) due to the presence of amide groups;
- After they were immersed in sodium chloride solution, the sodium ion content in the surface and subsurface of the two fibers was determined through X-ray spectroscopy elemental analysis, and finally, it is inferred that water can enter deeper into PPTA fibers, so under freeze-thaw cycles, it will cause greater damage to PPTA fibers.
Reference:
- Sawyer LC and Jaffe M. The structure of thermotropic copolyesters. Journal of Materials Science 1986; 21: 1897-1913.
- Sawyer L, Chen R, Jamieson M, et al. The fibrillar hierarchy in liquid crystalline polymers. Journal of materials science 1993; 28: 225-238.
- Wissbrun KF. Observations on the melt rheology of thermotropic aromatic polyesters. British Polymer Journal 1980; 12: 163-169.
- Wissbrun KF. A model for domain flow of liquid-crystal polymers. Faraday discussions of the Chemical Society 1985; 79: 161-173.
- Blundell D, Chivers R, Curson A, et al. The relationship of chain linearity of aromatic liquid-crystal polyesters to molecular orientation and stiffness of mouldings. Polymer 1988; 29: 1459-1467.
- Langelaan H and de Boer AP. Crystallization of thermotropic liquid crystalline HBA/HNA copolymers. Polymer 1996; 37: 5667-5680.
- Zhong Y. Morphology of thermotropic longitudinal polymer liquid crystals. Mechanical and Thermophysical Properties of Polymer Liquid Crystals 1998: 101-123.
- Sloan F. Liquid crystal aromatic polyester-arylate (LCP) fibers: Structure, properties, and applications. Structure and Properties of High-Performance Fibers. Elsevier, 2017, pp.113-140.

Reviewer 4 Report
Manuscript ID: polymers-2255387
Title: Implication of freeze-thaw erosion and mechanism analysis of 2 the high-performance aromatic liquid crystal fibers
The paper is interesting and draws attention to the application of two polymers in extreme exterior conditions.
Reviewer’s comments:
- The abstract too long. It should be shortened.
- A DSC (differential scanning calorimetry) analysis should be presented with the temperature domains where the studied materials have liquid crystal properties, and clear explanations. Very probable, at the low temperatures where all the experiments presented in this paper are conducted, the chosen materials are not in liquid crystal phase. Consequently, the title should be changed.
- English language: for example: r21: “ chemical structure of them”, r.99: “Another researchers have also proved”
- The whole paper should be carefully revised.
My recommendation is: Major revision of the manuscript.
Author Response
Dear Editor and Reviewers,
Thank you for your suggestion for our manuscript (polymers-2255387). And we have amended our manuscript and responded as follows. The change in the manuscript was made in RED (in the Revised manuscript with changes marked).
Response to the Reviewer #4’s Comments point by point:
------------------------------------------
Reviewer’s comment 1:
The abstract too long. It should be shortened.
Response:
Thank you for your suggestion.
According to the principle of the purpose, method, result and conclusion that the abstract must provide, this paper has two purposes: the first is to prove that the mechanical properties of TLCP fiber are superior; the second is to explore its mechanism. I have reduced more than 100 words. The revised abstract was as below:
In order to the demand for high-performance fibers for high-latitude ocean exploration and development, this paper selects representative products of high-performance liquid crystal fibers: thermotropic liquid crystal polymer (TLCP) fibers and poly p-phenylene terephthalamide (PPTA) fibers. Through a series of freeze-thaw (F-T) experiments for simulating a real chilly marine environment, then the retention of mechanical properties of these two kinds of fibers is tested. The mechanical properties of PPTA fibers decreases by about 25%, and the mechanical properties of TLCP fibers is almost no loss, indicating that TLCP fibers are better suited to cope with the operating standards of the severe frigid marine environment. Subsequently, the Fourier transformed infrared (FT-IR) in combination with attenuated total reflection (ATR) accessory analysis and the degree of crystal orientation measured by two-dimension wide-angle X-ray diffraction (2D-WAXD) confirm that no changes in their chemical and aggregated structure after experiencing the F-T cycles. However, as observed by scanning electron microscope (SEM), there are various extent of microcracks on the surface of PPTA fibers, but they do not appear on the surface of TLCP fibers. Certainly, before that, due to the difference in their chemical structure, we tested their Yang-Laplace contact angle (YLCA) and water absorption, the facts are preliminarily judged that PPTA fibers will absorb more moisture. Finally, the sodium ion solution is used to infiltrate the two fibers, and the elemental analysis of the fiber surface and subsurface sodium ions is obtained by Energy-dispersive X-ray spectroscopy (EDS), which proved that the microfibrillar structure of PPTA fibers is seriously damaged and deeply explored the Mechanism of this phenomenon.
Reviewer’s comment 2:
A DSC (differential scanning calorimetry) analysis should be presented with the temperature domains where the studied materials have liquid crystal properties, and clear explanations. Very probable, at the low temperatures where all the experiments presented in this paper are conducted, the chosen materials are not in liquid crystal phase. Consequently, the title should be changed.
Response:
Thanks again for your suggestion. After careful consideration, I think that PPTA fiber can only be dissolved because its decomposition temperature is lower than the melting temperature, so DSC cannot be used to test its liquid crystal properties; and the DSC curve of TLCP fiber is tested, as shown in the Fig 1, there is the only one melting peak, and it does not seem to clearly indicate its liquid crystal properties; maybe in the future research on thermal properties, I will definitely present such a spectrogram.
Since it is research under low temperature conditions, the title of the article is also in line with this meaning, we suggest the title of this manuscript needn’t be changed.
![]() |
|
Fig 1. DSC trace of TCLP fibers used in this article as a function of heating temperature |
Reviewer’s comment 3:
English language: for example: r21: “chemical structure of them”, r.99: “Another researchers have also proved”
Response:
I will pay more attention to avoid these mistakes, and thanks for your correction. I have made a revision, the problem of r.21 has been changed to “their chemical structure”; the problem of r.99, I have rephrased as follows: “Kevlar® solution spinning contains a small amount of sulfuric acid solvent, which causes the finished fiber to be catalyzed and hydrolyzed under high humidity, resulting in partial degradation and loss of mechanical properties”.
Reviewer’s comment 4:
The whole paper should be carefully revised.
Response:
Thank you for your reminding, I have carefully checked and corrected. And the changes were marked in red.

Round 2
Reviewer 4 Report
The authors did not answer satisfactory to my observations.
At those low temperatures, it very unlikely that the material is in liquid crystal state.
My conclusion about this paper is Reject and encourage resubmission.